# EQUIVARIANCE-AWARE ARCHITECTURAL OPTIMIZATION OF NEURAL NETWORKS

**Kaitlin Maile**
IRIT, University of Toulouse
kaitlin.maile@irit.fr

**Dennis G. Wilson**
ISAE-SUPAERO, University of Toulouse
dennis.wilson@isae-supaero.fr

**Patrick Forré**
University of Amsterdam
p.d.forre@uva.nl

## ABSTRACT

Incorporating equivariance to symmetry groups as a constraint during neural network training can improve performance and generalization for tasks exhibiting those symmetries, but such symmetries are often not perfectly nor explicitly present. This motivates algorithmically optimizing the architectural constraints imposed by equivariance. We propose the equivariance relaxation morphism, which preserves functionality while reparametrizing a group equivariant layer to operate with equivariance constraints on a subgroup, as well as the $[G]$-mixed equivariant layer, which mixes layers constrained to different groups to enable within-layer equivariance optimization. We further present evolutionary and differentiable neural architecture search (NAS) algorithms that utilize these mechanisms respectively for equivariance-aware architectural optimization. Experiments across a variety of datasets show the benefit of dynamically constrained equivariance to find effective architectures with approximate equivariance.

## 1 INTRODUCTION

Constraining neural networks to be equivariant to symmetry groups present in the data can improve their task performance, efficiency, and generalization capabilities (Bronstein et al., 2021), as shown by translation-equivariant convolutional neural networks (Fukushima & Miyake, 1982; LeCun et al., 1989) for image-based tasks (LeCun et al., 1998). Seminal works have developed general theories and architectures for equivariance in neural networks, providing a blueprint for equivariant operations on complex structured data (Cohen & Welling, 2016; Ravanbakhsh et al., 2017; Kondor & Trivedi, 2018; Weiler et al., 2021). However, these works design model constraints based on an explicit equivariance property. Furthermore, their architectural assumption of full equivariance in every layer may be overly constraining; e.g., in handwritten digit recognition, full equivariance to 180° rotation may lead to misclassifying samples of "6" and "9". Weiler & Cesa (2019) found that local equivariance from a final subgroup convolutional layer improves performance over full equivariance. If appropriate equivariance constraints are instead learned, the benefits of equivariance could extend to applications where the data may have unknown or imperfect symmetries.

Learning approximate equivariance has been recently approached via novel layer operations (Wang et al., 2022; Finzi et al., 2021; Zhou et al., 2020; Yeh et al., 2022; Basu et al., 2021). Separately, the field of neural architecture search (NAS) aims to optimize full neural network architectures (Zoph & Le, 2017; Real et al., 2017; Elsken et al., 2017; Liu et al., 2018; Lu et al., 2019). Existing NAS methods have not yet explicitly optimized equivariance, although partial or soft equivariant approaches like Romero & Lohit (2022) and van der Ouderaa et al. (2022) approach custom equivariant architectures. An important aspect of NAS is network morphisms: function-preserving architectural changes (Wei et al., 2016) which can be used during training to change the loss landscape and gradient descent trajectory while immediately maintaining the current functionality and loss value (Maile et al., 2022). Developing tools for searching over a space of architectural representations of equivariance would permit NAS algorithms to be applied towards architectural optimization of equivariance.

**Contributions** First, we present two mechanisms towards equivariance-aware architectural optimization. The *equivariance relaxation morphism* for group convolutional layers partially expands the representation and parameters of the layer to enable less constrained learning with a prior on symmetry. The $[G]$-*mixed equivariant layer* parametrizes a layer as a weighted sum of layers equivariant to different groups, permitting the learning of architectural weighting parameters.

Second, we implement these concepts within two algorithms for architectural optimization of partially-equivariant networks. Evolutionary Equivariance-Aware NAS (EquiNAS$_E$) utilizes the equivariance relaxation morphism in a greedy evolutionary algorithm, dynamically relaxing constraints throughout the training process. Differentiable Equivariance-Aware NAS (EquiNAS$_D$) implements $[G]$-mixed equivariant layers throughout a network to learn the appropriate approximate equivariance of each layer, in addition to their optimized weights, during training.

Finally, we analyze the proposed mechanisms via their respective NAS approaches in multiple image classification tasks, investigating how the dynamically learned approximate equivariance affects training and performance over baseline models and other approaches.

## 2 RELATED WORKS

**Approximate equivariance** Although no other works on approximate equivariance explicitly study architectural optimization, some approaches are architectural in nature. We compare our contributions with the most conceptually similar works to our knowledge.

The main contributions of Basu et al. (2021) and Agrawal & Ostrowski (2022) are similar to our proposed equivariant relaxation morphism. Basu et al. (2021) also utilizes subgroup decomposition but instead algorithmically builds up equivariances from smaller groups, while our work focuses on relaxing existing constraints. Agrawal & Ostrowski (2022) presents theoretical contributions towards network morphisms for group-invariant shallow neural networks: in comparison, our work focuses on deep group convolutional architectures and implements the morphism in a NAS algorithm.

The main contributions of Wang et al. (2022) and Finzi et al. (2021) are similar to our proposed $[G]$-mixed equivariant layer. Wang et al. (2022) also uses a weighted sum of filters, but uses the same group for each filter and defines the weights over the domain of group elements. Finzi et al. (2021) uses an equivariant layer in parallel to a linear layer with weighted regularization, thus only using two layers in parallel and weighting them by regularization rather than parametrization. Mouli & Ribeiro (2021) also progressively relaxes equivariance constraints, but with regularized rather than parametrized constraints.

In more diverse approaches, Zhou et al. (2020) and Yeh et al. (2022) represent symmetry-inducing weight sharing via learnable matrices. Romero & Lohit (2022) and van der Ouderaa et al. (2022) learn partial or soft equivariances for each layer.

**Neural architecture search** Neural architecture search (NAS) aims to optimize both the architecture and its parameters for a given task. Liu et al. (2018) approaches this difficult bi-level optimization by creating a large super-network containing all possible elements and continuously relaxing the discrete architectural parameters to enable search by gradient descent. Other NAS approaches include evolutionary algorithms (Real et al., 2017; Lu et al., 2019; Elsken et al., 2017) and reinforcement learning (Zoph & Le, 2017), which search over discretely represented architectures.

## 3 BACKGROUND

We assume familiarity with group theory (see Appendix A.1). For discrete group $G$, the $l^{\text{th}}$ $G$-equivariant group convolutional layer (Cohen & Welling, 2016) of a group convolutional neural network (G-CNN) convolves[1] the feature map $f\colon G \to \mathbb{R}^{C_{l-1}}$ from the previous layer with a filter with kernel size $k$ represented as learnable parameters $\psi\colon G \to \mathbb{R}^{C_l \times C_{l-1}}$. For each output channel

---

[1] We identify the correlation and convolution operators as they only differ where the inverse group element is placed and refer to both as "convolution" throughout this work.

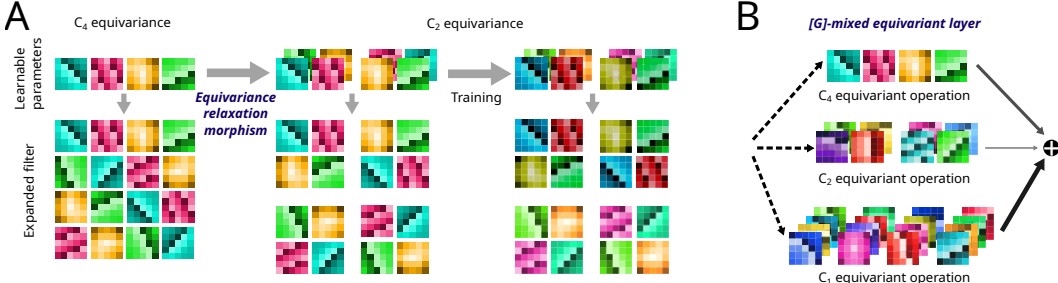

Figure 1: Visualizing the equivariance relaxation morphism and the $[G]$-mixed equivariant layer, using the $C_4$ group. In (A), the learnable parameters of a $C_4$-equivariant convolutional layer are expanded using each group element action so the expanded filter can be used in a standard convolutional layer. The equivariance relaxation morphism reparametrizes the layer to be architecturally constrained to $C_2$ equivariance, initialized to functional $C_4$ equivariance. In (B), convolutional operations equivariant to subgroups of $C_4$ are summed with learnable architectural weighting parameters.

$d \in [C_l]$, where $[C] := \{1, \ldots, C\}$, and group element $g \in G$, the layer's output is defined as:

$$[f \star_G \psi]_d(g) = \sum_{h \in G} \sum_{c=1}^{C_{l-1}} f_c(h)\psi_{d,c}(g^{-1}h). \tag{1}$$

The first layer is a special case: the input to the network needs to be lifted via this operation such that the output feature map of this layer has a domain of $G$. In the case of image data, an image $x$ with $C$ channels may be interpreted as a function $x \colon \mathbb{Z}^2 \to \mathbb{R}^C$ mapping each pixel in coordinate space to a real number for each channel, where the $c^{\text{th}}$ channel of $x$ is referred to as $x_c$. The input is $x \colon \mathbb{Z}^2 \to \mathbb{R}^{C_0}$, so the layer is instead a *lifting* convolution:

$$[x \star_G \psi]_d(g) = \sum_{y \in \mathbb{Z}^2} \sum_{c=1}^{C_0} x_c(y)\psi_{d,c}(g^{-1}y). \tag{2}$$

We present our contributions in the group convolutional layer case, although similar claims apply for the lifting convolutional layer case.

## 4 Towards Architectural Optimization over Subgroups

We propose two mechanisms to enable search over subgroups: the equivariance relaxation morphism and a $[G]$-mixed equivariant layer. The proposed morphism, depicted in Figure 1(A) and described in Section 4.1, changes the equivariance constraint from one group to another subgroup while preserving the learned weights of the initial group convolutional operator. The $[G]$-mixed equivariant layer, shown in Figure 1(B) and presented in Section 4.2, allows for a single layer to represent equivariance to multiple subgroups through a weighted sum.

### 4.1 Equivariance Relaxation Morphism

The *equivariance relaxation morphism* reparametrizes a $G$-equivariant group (or lifting) convolutional layer to operate over a subgroup of $G$, partially removing weight-sharing constraints from the parameter space while maintaining the functionality of the layer.

Let $G' \leq G$ be a subgroup of $G$ such that $G' \setminus G$ is finite. Let $R$ be a system of representatives of the left quotient (including the identity element), so that $G' \setminus G = \{G'r \mid r \in R\}$, where $G'r := \{g'r \mid g' \in G'\}$. Given a $G$-equivariant group convolutional layer with feature map $f$ and filter $\psi$, we define the relaxed feature map $\tilde{f} \colon G' \to \mathbb{R}^{(C_{l-1} \times |R|)}$ and relaxed filter $\tilde{\psi} \colon G' \to \mathbb{R}^{(C_l \times |R|) \times (C_{l-1} \times |R|)}$ as follows. For $c \in [C_{l-1}]$, $s,t \in R$, $d \in [C_l]$:

$$\tilde{f}_{(c,s)}(g') := f_c(g's), \tag{3}$$

$$\tilde{\psi}_{(d,t),(c,s)}(g') := \psi_{d,c}(t^{-1}g's). \tag{4}$$

We define the *equivariance relaxation morphism* from $G$ to $G'$ as the reparametrization of $\psi$ as $\tilde{\psi}$ (Eq. 4) and reshaping of $f$ as $\tilde{f}$ (Eq. 3). We will show that the new layer, $[\tilde{f} \star_{G'} \tilde{\psi}]_{(d,t)}(g')$, is equivalent to $[f \star_G \psi]_d(g't)$ down to reshaping. Since the mapping $G' \times R \to G$, $(g', t) \mapsto g't$, is bijective, every $g$ can uniquely be written as $g = g't$ with $g' \in G'$ and $t \in R$. For $g \in G$, $G'g \in G' \setminus G$ has a unique representative $t \in R$ with $G'g = G't$, and $g' := gt^{-1} \in G'$. Similarly, $h \in G$ may be written as $h = h's$ with unique $h' \in G'$ and $s \in R$. With these preliminaries, we get:

$$[f \star_G \psi]_d(g't) = [f \star_G \psi]_d(g) \tag{5}$$

$$= \sum_{h \in G} \sum_{c=1}^{C_{l-1}} f_c(h) \psi_{d,c}(g^{-1}h), \tag{6}$$

$$= \sum_{h' \in G'} \sum_{s \in R} \sum_{c=1}^{C_{l-1}} f_c(h's) \psi_{d,c}(t^{-1}g'^{-1}h's), \tag{7}$$

$$= \sum_{h' \in G'} \sum_{c=1}^{C_{l-1}} \sum_{s \in R} \tilde{f}_{(c,s)}(h') \tilde{\psi}_{(d,t),(c,s)}(g'^{-1}h'), \tag{8}$$

$$= \left[\tilde{f} \star_{G'} \tilde{\psi}\right]_{(d,t)}(g'), \tag{9}$$

which shows the claim. Thus, the convolution of $\tilde{f}$ with $\tilde{\psi}$ is equivariant to $G$ but parametrized as a $G'$-equivariant group convolutional layer, where the representatives are expanded into independent channels. This morphism can be viewed as initializing a $G'$-equivariant layer with a pre-trained prior of equivariance to $G$, maintaining any previous training.

Standard convolutional layers are a special case of group-equivariant layers, where the group is translational symmetry over pixel space. Regular group convolutions are often implemented by relaxation to the translational symmetry group by expanding the filter via the appropriate group actions, allowing a standard convolution implementation from a deep learning library to be used. The equivariance relaxation morphism generalizes this concept to any subgroup. This, as well as how the equivariance relaxation morphism is implemented, is discussed further in Appendix B.

## 4.2 $[G]$-MIXED EQUIVARIANT LAYER

Towards learning equivariance, we additionally propose partial equivariance via a mixture of layers, each constrained to equivariance to different groups, applied in parallel to the same input then combined via a weighted sum. The equivariance relaxation morphism provides a mapping of group elements between a group and a subgroup. For a set of groups $[G]$, such as a subgroup lattice of some group $G$, we define a $[G]$-*mixed equivariant layer* as:

$$\left[f \hat{\star}_{[G]} [\psi]\right]_{(d,t)}(g) = \sum_{G \in [G]} z_G \left[f \star_{G'} \widetilde{\psi^G}\right]_{(d,t)}(g) \tag{10}$$

$$= \left[f \star_{G'} \sum_{G \in [G]} z_G \widetilde{\psi^G}\right]_{(d,t)}(g), \tag{11}$$

where each element $z_G$ of $[z] := \{z_G | G \in [G]\}$ is an architectural weighting parameter such that $\sum_{G \in [G]} z_G = 1$, $G'$ is a subgroup of all groups in $[G]$, each element $\psi^G$ of $[\psi]$ is a filter with a domain of $G$, and $\widetilde{\psi^G}$ is the transformation of $\psi^G$ from a domain of $G$ to $G'$ as defined in Equation 4. Thus, the layer is parametrized by $[\psi]$ and $[z]$, computing a weighted sum of operations that are equivariant to different groups of $[G]$. The layer may be equivalently computed by convolution of the input with the weighted sum of transformed filters, shown in Equation 11. We provide further implementation details in Appendix B.

## 5 EQUIVARIANCE-AWARE NEURAL ARCHITECTURE ALGORITHMS

We present two NAS methods that utilize the presented mechanisms for discovering appropriate equivariance during training: Evolutionary Equivariance-Aware NAS (EquiNAS$_E$) and Differen-

---

**Algorithm 1** Evolutionary equivariance-aware neural architecture search.

---

**procedure** EQUINAS$_E$(Initial symmetry group $G$)
 Initialize population with a $G$-equivariant group convolutional network.
 **for** each generation **do**
  **for** each network in population **do**
   Add children of network with relaxed equivariance constraints into population.
  **for** each network in population **do**
   Partially train network on dataset.
  Select Pareto-efficient and high accuracy networks as new population.
 **return** population

---

tiable Equivariance-Aware NAS (EquiNAS$_D$). Both methods optimize an architecture while learning weights, yielding a final trained network adapted to equivariances present in the training data. However, they differ in NAS paradigm and approximate equivariance representation: EquiNAS$_E$, in Section 5.1, searches for networks composed of layers each fully equivariant to possibly different groups, while EquiNAS$_D$, in Section 5.2, searches for smooth mixtures of equivariant layers.

## 5.1 EVOLUTIONARY EQUIVARIANCE-AWARE NAS

Towards finding the optimal full equivariance per layer, the equivariance relaxation morphism presented in Section 4.1 is applied as the genetic operator in an evolutionary hill-climbing algorithm. The Evolutionary Equivariance-Aware NAS (EquiNAS$_E$) algorithm, given in Algorithm 1, is similar to other evolutionary NAS methods such as Elsken et al. (2017) with pareto selection as in Falanti et al. (2022). A population of networks, which starts with an individual with all layers equivariant to the largest possible group, undergoes mutation via equivariance relaxation and selection based on accuracy and parameter count to optimize neural architecture while learning network parameters. See Appendix A.2 for further background on evolutionary NAS.

In each generation, candidate networks are evaluated based on maximizing validation accuracy and minimizing parameter count: the pareto-dominant individuals with highest accuracy are kept, then additional high-accuracy individuals are added if necessary until the desired parent population size is reached. Offspring are generated from each parent separately by mutation using the relaxation morphism. This preserves the weights of the parametrized equivariance during mutation, allowing for the continuous training of networks over evolution by inheritance from parent individuals. Specifically, mutation reduces a single layer's parametrized equivariance to a subgroup within the constraint that each layer has parametrized equivariance to a subgroup of all preceding layers. This constraint yields local equivariance properties for the network, as shown in Weiler & Cesa (2019) and Elsayed et al. (2020) to be empirically favorable in image classification tasks. The resulting individuals are each trained independently for a given training time, and then this process repeats.

The second objective of minimizing parameter count is intended to advance efficient networks, such as those with large symmetry groups. Accuracy-based selection alone would necessarily prefer larger networks as mutation via the equivariance relaxation morphism results in two networks with identical performance but different size, the relaxed network having more parameters, until training; potentially short-term increases in validation accuracy after training would then result in the selection of individuals with more parameters. Thus, the proposed strategy of selecting both pareto-dominant and high-accuracy individuals is intended to maintain a diverse yet efficient population without succumbing to overly greedy selections too early.

## 5.2 DIFFERENTIABLE EQUIVARIANCE-AWARE NAS

In a contrasting paradigm, the $[G]$-mixed equivariant layer presented in Section 4.2 allows for smoothly searching across a spectrum of equivariance for each layer via a differentiable NAS algorithm. Our Differentiable Equivariance-Aware NAS (EquiNAS$_D$) algorithm, defined in Algorithm 2, is inspired by DARTS (Liu et al., 2018) with significant changes detailed in the following paragraphs. EquiNAS$_D$ simplifies the bilevel optimization of the architecture weighting parameters $Z$ and filter weights $\Psi$ into alternating independent updates, computing the gradient update for $Z$ with the current, rather than optimal, $\Psi$ for the current architecture encoded by $Z$, to boost search efficiency with minimal performance loss compared to higher order approximations (Liu et al., 2018).

---

**Algorithm 2** Differentiable equivariance-aware neural architecture search.

---

**procedure** EQUINAS$_D$(Set of groups $[G]$)
    Initialize network with $[G]$-mixed equivariant layers, parametrized by $\Psi$ and $Z$.
    **while** not converged **do**
        Update $Z$ by $\nabla_Z \mathcal{L}(\Psi, Z)$.
        Update $\Psi$ by $\nabla_\Psi \mathcal{L}(\Psi, Z)$.
    **return** trained network

---

In most differentiable NAS search spaces, the desired output architecture is discretized to select a subset of architectural options within constraints, then the weights are re-initialized and trained within the static architecture. In our formulation, this is not necessary as any mixed operation can be equivalently expressed as a single layer equivariant to any group $G'$ that is a common subgroup to all groups of the mixed operation (Eq. 11): in our experimental case, this is a standard translation-equivariant convolutional layer, so the final model can be equivalently expressed as a standard convolutional model with encoded partial equivariance. Thus, the final optimized architecture and trained weights are output from the single search process. We explore the standard NAS paradigm, where weights are reinitialized and trained in the final static architecture, in Appendix D.

In order to enforce that the scaling of each filter does not confound the architecture weighting parameters, we use the weight normalizing reparametrization (Salimans & Kingma, 2016) and do not update the scalar norm parameter of each filter after initialization.

We do not use disjoint datasets for updating $\Psi$ and $Z$, but rather draw one batch for $\Psi$ and another for $Z$ independently and randomly from the same training split. This allows for a standard dataset split and to use the validation set for hyperparameter tuning.

These two NAS approaches present adaptations of two standard types of NAS, evolutionary and differentiable, to the search for optimal partial equivariance. We next study empirically the two EquiNAS methods on three datasets, one with known rotational symmetry and two with unknown but visually significant rotational and reflectional symmetry.

## 6 EXPERIMENTS

We focus on the regular representation of groups and show experiments with reflectional and up to 4-fold rotational symmetry groups applied to image classification tasks. Examples of symmetry groups acting on pixel space, which corresponds to $\mathbb{Z}^2$, include $T(2)$, which consists of discrete translations in both dimensions; the cyclical groups $C_n$, which consist of $n$-fold rotations; and the dihedral groups $D_n$, which consist of reflections with $n$-fold rotations, where $n \in \{1, 2, 4\}$ for exact symmetry without interpolation. The $p4$ group consists of discrete translations and multiples of $90°$ rotations and may be represented as $T(2) \rtimes C_4$. The $p4m$ group consists of discrete translations, reflections, and multiples of $90°$ rotations and may be represented as $T(2) \rtimes D_4$. As standard convolutional layers are already equivariant to $T(2)$, we refer to layers also equivariant to $n$-fold rotations with or without reflections as $D_n$ or $C_n$-equivariant, respectively. So, a $C_1$ equivariant convolutional layer is a standard translation-equivariant convolutional layer. We use $\{C_1, D_1, C_2, D_2, C_4, D_4\}$ as the set of potential groups for mutation in EquiNAS$_E$ and as $[G]$ in EquiNAS$_D$.

We present experiments on image classification for a variety of datasets. The Rotated MNIST dataset (Larochelle et al., 2007, RotMNIST) is a version of the MNIST handwritten digit dataset but with the images rotated by any angle. This task serves as a simple investigational study with known symmetry, while the following two tasks are more realistic and complex. The Galaxy10 DECals dataset (Leung & Bovy, 2019, Galaxy10) contains galaxy images in 10 broad categories. The ISIC 2019 dataset (Codella et al., 2018; Tschandl et al., 2018; Combalia et al., 2019, ISIC) contains dermascopic images of 8 types of skin cancer plus a null class. For Galaxy10 and ISIC, we downsample the images to $64 \times 64$ due to computational constraints, which adds notable difficulty to the tasks. These tasks exhibit varying levels of rotational and reflectional symmetry, motivating architectural optimization to determine the most effective application of equivariance constraints.

Across all experiments, the architectures are designed to have consistent channel dimensions once expanded to a standard translation-equivariant convolutional layer for each layer across models. Thus, constrained equivariance to a larger symmetry group results in fewer learnable parameters. A

| Method | RotMNIST | Galaxy10 | ISIC |
|---|---|---|---|
| EquiNAS$_E$ | **1.78 ± 0.04** | **20.3 ± 0.9** | **31.0 ± 0.4** |
| RPP (Finzi et al., 2021) | 2.18 ± 0.04 | **24.3 ± 2.8** | 32.2 ± 1.7 |
| $D_4$ baseline | **1.78 ± 0.11** | 50.8 ± 17.0 | 32.1 ± 2.4 |
| $C_4$ baseline | **1.64 ± 0.22** | 29.6 ± 5.5 | 32.9 ± 1.0 |
| $C_1$ baseline | 5.02 ± 1.15 | 31.6 ± 4.8 | 33.2 ± 1.5 |
| $C_4$ (prior: $D_4$) | 1.93 ± 0.05 | 27.8 ± 5.3 | 31.9 ± 1.5 |
| $C_1$ (prior: $D_4$) | 3.40 ± 0.07 | 25.9 ± 2.3 | **31.4 ± 2.6** |
| $C_1$ (prior: $C_4$) | 2.96 ± 0.05 | 30.7 ± 7.1 | 32.5 ± 1.1 |

Table 1: Test error (lower is better) in percent of incorrect classifications across tasks and approaches. Statistics are aggregated over the final selected population of 5 individuals for EquiNAS$_E$ and across 5 random seeds for all other methods. The **best** and **second best** average errors per task are highlighted. See Figures 5-6 in Appendix D for individual results and additional experiments.

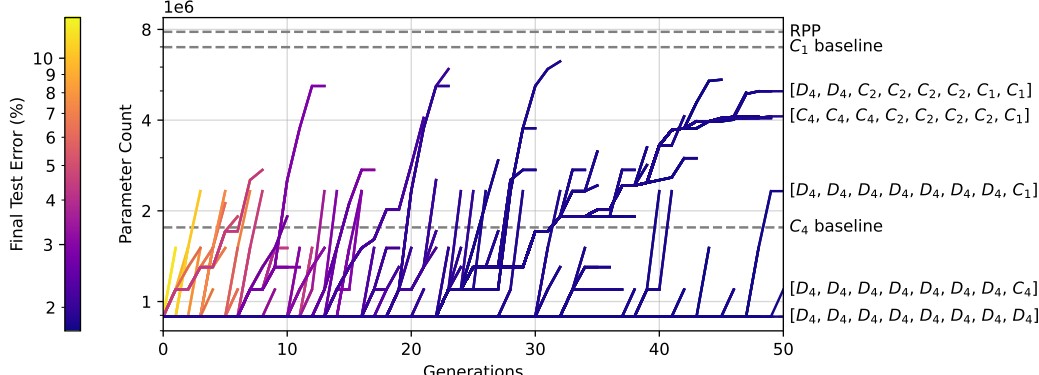

Figure 2: Historical parameter counts of all selected individuals for EquiNAS$_E$ on RotMNIST. The architectures of the final selected population are labeled. Each history is colored by the final test accuracy, measured of each individual upon removal. For other tasks, see Figure 7 in Appendix D.

layer constrained to $C_4$ equivariance has $|C_4 \setminus D_4| = 2$ times as many independent channels and as many parameters as a layer constrained to $D_4$ equivariance. This is a notably different paradigm than other works that equate parameter counts across architectures with different equivariance properties.

As baseline comparisons, we train and test G-CNNs with static architectures. In addition to the static baselines, we re-implement the residual pathway priors (RPP) approach by Finzi et al. (2021) as a $C_1$ equivariant layer with regularization in parallel with a $D_4$ equivariant convolutional layer.

Further experiment details such as architecture details and other hyperparameters are in Appendix C. For each paradigm of experiments, we present results in the following subsections, with general discussion in Section 7. Additional ablation and random search baselines are in Appendix D.

## 6.1 EVOLUTIONARY EQUIVARIANCE-AWARE NAS

The classification test errors are listed in Table 1. The advantages of equivariance search methods are most apparent in the Galaxy10 benchmark. While EquiNAS$_E$ outperforms most baselines on RotMNIST and all baselines on ISIC, it has similar performance and some final architectures to the $D_4$ baseline for both tasks. However, the $D_4$ baseline fails at the Galaxy10 task, demonstrating that the same equivariant architecture can not always be naively applied. Both search methods, EquiNAS$_E$ and RPP, outperform all baseline models on Galaxy10, and by a large margin for EquiNAS$_E$.

The evolutionary progress on RotMNIST is shown in Figure 2: the selected population maintains a fully equivariant network in every generation. The final selected population originates from two main lineages, one staying fully equivariant until the last generations and the other diverging from

| Method | RotMNIST | Galaxy10 | ISIC |
|---|---|---|---|
| EquiNAS$_D$ | **2.29 ± 0.27** | **21.8 ± 1.2** | 32.8 ± 0.6 |
| RPP (Finzi et al., 2021) | 2.89 ± 0.27 | 22.0 ± 1.8 | **31.5 ± 0.9** |
| $D_4$ Baseline | 2.97 ± 1.50 | 22.5 ± 2.0 | **32.0 ± 1.0** |
| $C_4$ Baseline | **2.43 ± 0.54** | 22.2 ± 2.4 | 32.8 ± 1.0 |
| $C_1$ Baseline | 3.97 ± 0.75 | 26.5 ± 1.5 | 32.9 ± 3.1 |

Table 2: Test error (lower is better) in percent of incorrect classifications across tasks and approaches. The **best** and **second best** average errors per task are highlighted. See Figures 8-9 in Appendix D for individual results and additional experiments.

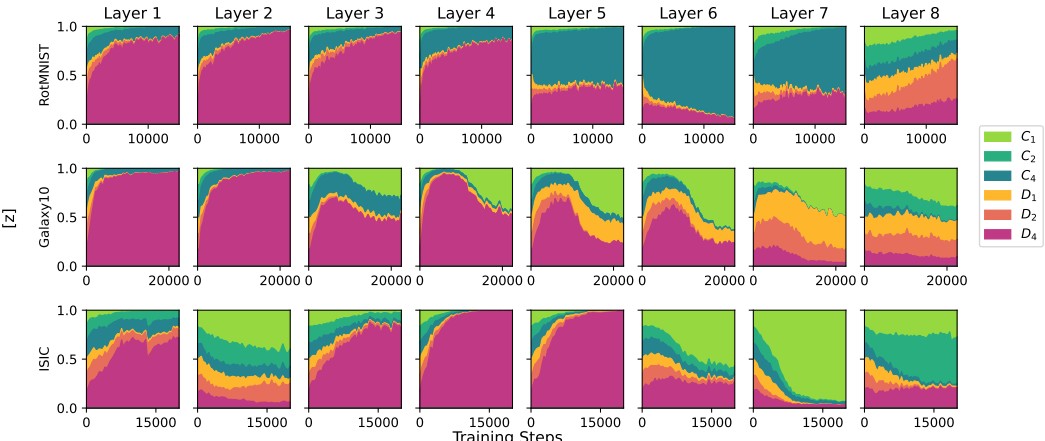

Figure 3: Architecture weighting parameters by layer for one selected trial from RotMNIST, Galaxy10, and ISIC. For other trials, see Figures 10-12 in Appendix D.

the fully equivariant network midway through, showing that training with dynamically constrained parametrizations can produce performant models.

In addition to the normally initialized static baselines, we also train and test baselines that are initialized with priors to larger symmetry groups. These are implemented by initializing all layers to be constrained to the prior symmetry group, then using the equivariance relaxation morphism on each layer. EquiNAS$_E$ searches for relaxation schedules that yield trained priors on equivariance, while these additional baselines yield untrained priors. The results in Table 1 show that, the $C_1$-equivariant networks generally improve with either equivariance prior, while the $C_4$ equivariant networks perform better with $D_4$ equivariance initialization only when the $D_4$ constrained baselines also work well. The untrained prior methods do not perform as well as EquiNAS$_E$ on RotMNIST, showing the benefit of investing some training time to the constrained equivariance. For the other tasks, the baselines with priors have better performances than their constrained baseline counterparts.

### 6.2 DIFFERENTIABLE EQUIVARIANCE-AWARE NAS

The classification test errors are listed in Table 2. EquiNAS$_D$ achieves better test accuracy than the other comparable methods on RotMNIST and Galaxy10. Due to differences in training protocol, only comparisons of relative rankings with Table 1 are possible: baseline methods accuracies follow similar ranking patterns, suggesting the benefit of general $C_4$ equivariance for RotMNIST and Galaxy10 and general $D_4$ equivariance, including RPP, for ISIC. In this training protocol notably with adaptive optimizers, the results are more consistent across methods and trials.

The dynamics of architecture weighting parameters for one exemplary trial per task are shown in Figure 3. The general trend of less constrained layers toward the output supports the conjecture of local equivariance being beneficial. However, this effect is less consistent for ISIC, the only task where EquiNAS$_D$ did not exceed baselines, possibly indicating less inherent symmetry. As seen in

Appendix D, the final mixing of architectures for ISIC included a high level of $C_1$, indicating that feature analysis outside of these symmetry groups is important for this benchmark.

Previous differentiable NAS works often used regularization of network size or even architecture weighting parameters themselves to encourage efficient architectures with a single highly weighted choice for each layer. However, our algorithm shows strong preference for a single, more equivariant and thus more expressive layer, notably to $D_4$ or $C_4$ equivariance, without such regularization. This may be due to the bilevel optimization dynamics: more constrained layers may be able to make more effective updates and thus become favorable compared to the lagging larger layers.

## 7    DISCUSSION

To our knowledge, this is the first work which proposes search methods for networks with dynamically constrained equivariance. Many NAS approaches separately search for an architecture and then reinitialize and retrain the weights, while our two proposed approaches find an optimal architecture with trained weights in a single process, notably with dynamically constrained weights. Gradient-based tuning (Maclaurin et al., 2015) has shown the benefit not only of optimizing hyperparameters but also of dynamically adjusting them during training (Lichtarge et al., 2022). Dynamically constrained weights can reap the benefits of specialization and generalization over the course of training.

Our two equivariance-aware NAS approaches have distinct approaches: EquiNAS$_E$ searches for architectures composed of discretely equivariant layers, while EquiNAS$_D$ searches for continuous mixtures of equivariance within each layer. The EquiNAS$_D$ algorithm avoids many known problems in differentiable NAS such as the discretization gap that occurs when searching over a continuous relaxation of a discrete architectural search space (Xie et al., 2021), such as that of EquiNAS$_E$. Towards searching for discretely equivariant layers using the $[G]$-mixed equivariant layer, proximal NAS algorithms use techniques such as projection (Yao et al., 2020) and straight-through estimation (Li et al., 2022) to avoid the discretization gap and thus may be effective for this application.

EquiNAS$_E$ is innately greedy: at each selection step, the population is evaluated by known current performance rather than unknown final performance, biased to architectures that train quickly. Networks with more equivariance constraints tend to learn faster, but equivariance relaxation may yield large gradients for newly unconstrained parameters and thus fast increases in performance. Further work could utilize metrics for final performance, such as proxies (White et al., 2022).

The theoretical and algorithmic contributions of this work are applicable beyond the image classification experiments presented to architectures with parametrized equivariance to any discrete group. We leave the extension to other group representations and domains as future work, such as the continuous case via careful analysis of the regular representation, still given $G' \setminus G$ is finite.

Our proposed equivariance-aware NAS problems can be practically applied to find effective models or architectures for datasets with hypothesizable symmetry. EquiNAS$_E$ may particularly work well on tasks that benefit from local equivariance, determined by analyzing the architecture weighting parameters from first applying the more efficient EquiNAS$_D$, as well as for finding good discrete architectures within which to retrain weights, based on the ablation and random comparisons. We thus recommend EquiNAS$_D$ for practical applications if the final model is not restricted to discrete equivariance, in which case it can be used to inform design decisions for applying EquiNAS$_E$.

Beyond NAS, the equivariance relaxation morphism could be used in other applications such as fine-tuning and distillation. Layers of a pre-trained equivariant network could be expanded via equivariance relaxation before fine-tuning on the same or a new task. Similarly, a network could be distilled to a wider architecture for additional performance benefits.

**Conclusion**    We present two mechanisms towards equivariance-aware architectural optimization, the equivariance relaxation morphism and the $[G]$-mixed equivariant layer, and two NAS algorithms that respectively implement these mechanisms evolutionarily as EquiNAS$_E$ and differentiably as EquiNAS$_D$. We investigate how dynamic equivariance achieved by these algorithms affects the training and performance of models across multiple image classification tasks of varying complexity and assumed symmetry, demonstrating that these techniques can search for performant architectures and weights even on noisy tasks. The proposed mechanisms and algorithms are extendable beyond vision tasks to any architecture with parametrized equivariance to any discrete group.

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

## A  ADDITIONAL BACKGROUND

### A.1  SYMMETRIES IN NEURAL NETWORKS

A symmetry of an object is a mapping of the object onto itself such that structure is preserved. A *symmetry group* $G$ is a set of such mappings along with a binary operation $\cdot\colon G \times G \to G$, known as the *group product*, that satisfies axioms for closure, associativity, the identity, and the inverse (Herstein, 2006). A group $G$ acts on a set $\mathcal{X}$ via the *group action* $.\colon G \times \mathcal{X} \to \mathcal{X},\ (g, x) \mapsto g.x$ that satisfies axioms for identity and compatibility: $\mathcal{X}$ is called a *G-space*.

Equivariance is the property of a mapping such that transformation of the input results in equivalent transformation of the output. Formally, a mapping $h\colon \mathcal{X} \to \mathcal{Z}$ between two $G$-spaces is $G$-equivariant if for all $g \in G$ and $x \in \mathcal{X}$ we have: $h(g.x) = g.h(x)$. For example, an image segmentation neural network should be $T(2)$-equivariant: shifting the input should result in the same shift in the output.

Invariance is a special case of equivariance, where the output of the function is completely independent of transformation of the input. Formally, a mapping $h\colon \mathcal{X} \to \mathcal{Z}$ is $G$-invariant if for all $g \in G$ and $x \in \mathcal{X}$ we have: $h(g.x) = h(x)$. For example, an image classification network should be $T(2)$-invariant: shifting the input should not change the output. Symmetries leave objects invariant.

For two groups $G$ and $H$ with group products $\cdot_G$ and $\cdot_H$ respectively where $H$ acts on $G$ with group action $.$, the (outer) semi-direct product $G \rtimes H$ of $H$ acting on $G$ is a group composed of the set of elements $G \times H$ with group product $(g, h) \cdot (g', h') = (g \cdot_G (h.g'), h \cdot_H h')$ and inverse $(g, h)^{-1} = (h^{-1}.g^{-1}, h^{-1})$.

A *subgroup* $H$ of $G$ is a nonempty subset with the same group product that also fulfills the group axioms. Then, $gH = \{g \cdot h | h \in H\}$ and $Hg = \{h \cdot g | h \in H\}$ denote the *left coset* and *right coset*, respectively, of $H$ with representative $g$.

### A.2  NEURAL ARCHITECTURE SEARCH

Evolutionary algorithms are optimization methods inspired by evolution in biology, where individuals in a population compete with their phenotypic traits in order to pass on their genotypic traits to offspring. The *population* is the current collection of individuals. Each *individual* is an instance of the object to be optimized and has a genotype that is decoded into a phenotype. In this case, each individual is a neural network, with a genotype that encodes the parametrized equivariance group of each convolutional layer, represented as a vector of integers. The individual continues training on the task before competing against other individuals to be selected as a parent to mutate to generate the next population. Each parent itself is kept for the next population, as well as each valid child that is generated via the equivariance relaxation morphism, such that they are functionally equivalent to their parent at initialization (although with a different architecture) and thus have the same fitness before training. Each individual in this population is partially trained, such that these children diverge from their siblings and parent, so that the next set of parents may be selected and this process repeats.

Pareto dominance can be used in multi-objective optimizations to select the next parent population. For a population of individuals each scored in $n$ objectives, an individual is *pareto-optimal* if no individual has at least one strictly better score for an objective without that of any other objectives being strictly worse. The *pareto front* is the set formed by all pareto-optimal individuals.

## B    IMPLEMENTATION DETAILS

**Group convolutional layers**    The implementation of regular group convolutional layers can be viewed as a special case of our proposed equivariance relaxation morphism. With the preliminaries given in Section 4.1 and the case of $G' = T(2)$, $\tilde{f}$ and $\tilde{\psi}$ are computed such that $\tilde{f}_{(c,s)}(g') := f_c(g's)$ and $\tilde{\psi}_{(d,t),(c,s)}(g') := \psi_{d,c}(t^{-1}g's)$ for each $g' \in T(2)$, $c \in [C_{l-1}]$, $s, t \in R$, and $d \in [C_l]$.

Let $S_G := |R|$. The learnable parameters of the $G_l$-equivariant $l^{\text{th}}$ layer with $C_l$ output channels, corresponding to $\psi$, are stored as a tensor of size $C_l \times C_{l-1} \times S_{G_l} \times K_l \times K_l$. The filter transformation expands this filter tensor by performing the action of each $r \in R$ on another copy of the tensor to expand its shape along a new dimension, resulting in a tensor of size $C_l \times S_{G_l} \times C_{l-1} \times S_{G_l} \times K_l \times K_l$, which is reshaped to $C_l S_{G_l} \times C_{l-1} S_{G_l} \times K_l \times K_l$. The input tensor to the $l^{\text{th}}$ layer, corresponding to $f$, is in the shape of $B \times C_{l-1} \times S_{G_l} \times H_{l-1} \times W_{l-1}$, which is reshaped to $B \times C_{l-1} S_{G_l} \times H_{l-1} \times W_{l-1}$ and convolved with the expanded filter. The output of shape $B \times C_l S_{G_l} \times H_l \times W_l$ is reshaped to $B \times C_l \times S_{G_l} \times H_l \times W_l$.

**Equivariance relaxation morphism**    To implement the equivariance relaxation morphism, the new filter tensor is initialized by applying Equation 4 such that result of applying the preceding filter transformation is equivalent. Our implementation of group actions relies on group channel indexing to represent the order of group elements: to ensure this is consistent before and after the morphism, the appropriate reordering of the output and input channels of the expanded filter are applied upon expansion. The new filter tensor has a shape of $C_l|R| \times C_{l-1}|R| \times S_{G_l}/|R| \times K_l \times K_l$. The $[G]$-mixed equivariant layer is built on top of this implementation, also using proper input and output channel reordering between layers to ensure correct mixing of group channels.

## C    EXPERIMENTAL DETAILS

**Architecture backbone**    For both EquiNAS$_E$ and EquiNAS$_D$ experiments, we use the same backbone architecture, such that the static baselines have the same architecture across experiments. The architectures have a lifting layer followed by 7 group convolutional layers, for a total of 8 convolutional layers. After 4 layers, the channel count doubles, from 16 to 32 for a $D_4$ equivariant layer and scaling up for smaller symmetry group equivariance constraints. An average pooling layer is placed after every other layer for all architectures and additionally after the fifth and seventh convolutional layers for Galaxy10 and ISIC. After the final group convolutional layer is a group-dimension average pooling followed by two linear layers to the output dimension. Every convolutional and linear layer except the output layer is immediately followed by a batchnorm then a ReLU.

**Hyperparameters**    The hyperparameters for each algorithm are selected such that baselines only differ by training time and optimizers. The learning rates were selected by grid search over baselines on RotMNIST. For all experiments in Sections 6.1, we use a simple SGD optimizer with learning rate $0.1$ to avoid confounding effects such as momentum during the morphism. For EquiNAS$_E$, the parent selection size is 5, the training time per generation is 0.5 epochs, and the number of generations is 50 for all tasks. Baselines were trained for the equivalent number of epochs. For all experiments in Section 6.2, we use separate Adam optimizers for $\Psi$ and $Z$, each with a learning rate of $0.01$ and otherwise default settings. The total training time is 100 epochs for RotMNIST and 50 epochs for Galaxy10 and ISIC. For RPP, we use a $C_1$-equivariant layer with an $L2$ regularization parameter of $1e-6$ in parallel with a $D_4$-equivariant layer without regularization.

For RotMNIST and MNIST, we use the standard training and test splits with a batch size of 64, reserving 10% of the training data as the validation set. For Galaxy10, we set aside 10% of the dataset as the test set, reserving 10% of the remaining training data as the validation set. For ISIC, we set aside 10% of the available training dataset as the test set, reserving 10% of the remaining data as the validation set and the rest as training data. For the latter two datasets, we resize the

images to $64 \times 64$ due to computational constraints and use a batchsize of 32. The validation sets were previously used for hyperparameter tuning: for experimental results, they are only used for the experiments in Section 6.1 as necessary for the EquiNAS$_E$ algorithm. No data augmentation is performed, although the datasets are normalized.

# D  ADDITIONAL EXPERIMENTS AND FIGURES

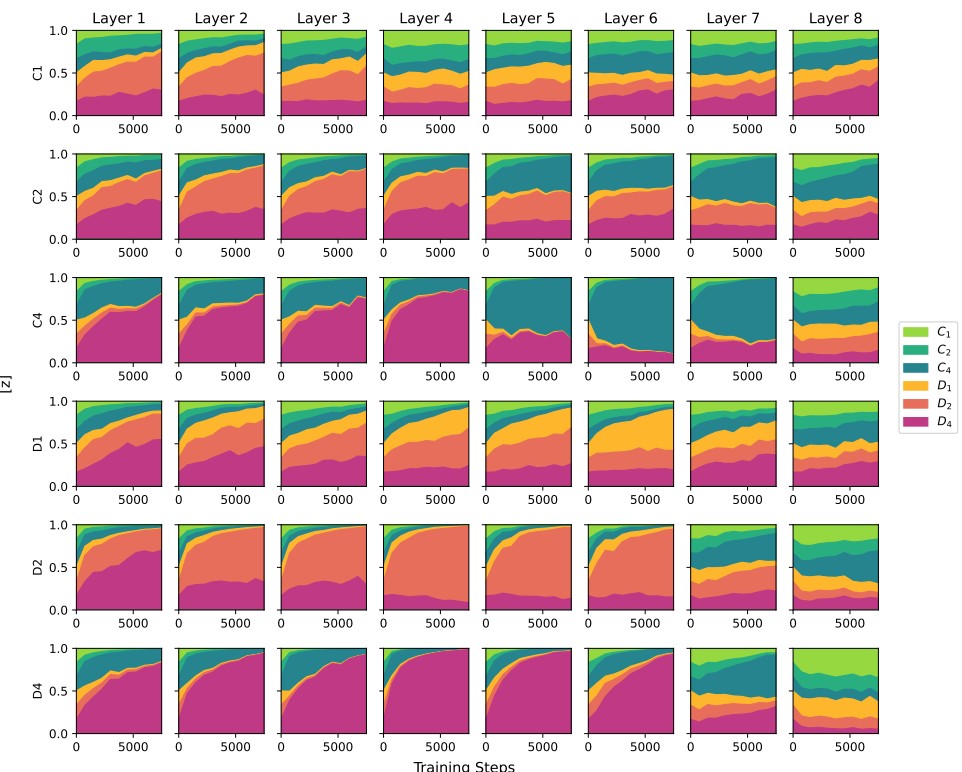

Figure 4: EquiNAS$_D$ on classification of six augmentations of MNIST, where each row is a trial on MNIST augmented to exhibit symmetry to the labeled group.

To explore the symmetry discovery of EquiNAS$_D$, we apply it to six augmentations of the MNIST dataset (LeCun et al., 1998), where each augmentation applies the group actions of each group in $\{C_1, C_2, C_4, D_1, D_2, D_4\}$ respectively. The resulting architecture dynamics of this experiment are shown in Figure 4, showing that less augmented versions still have some inherent symmetry, while more augmented versions induce stronger architectural changes towards layers that are equivariant to larger groups. Across all augmentations, earlier layers tended towards more constraints to equivariance.

As ablation studies and comparisons, we implement two kinds of random search for each NAS method. The first ablates smart architecture search: EquiNAS$_E$ Random Select works as described in Algorithm 1 but with random parent selection (instead of pareto-front selection) and EquiNAS$_D$ Random $Z$ works as described in Algorithm 2 but with random $Z$ updates (instead of gradient descent) by shuffling $Z$ gradients. The second is more akin to standard NAS random search: for the evolutionary paradigm, we train 30 randomly selected static architectures in the discrete architecture search space for the same training time and selecting the top 5 by validation accuracy, and for the differentiable paradigm, we train 25 randomly selected static architectures in the continuous architecture search space and selecting the top 5 by validation accuracy. 30 and 25 were respectively calculated to be approximately the same compute cost as the trials of EquiNAS$_E$ and EquiNAS$_D$. These are labeled as "Random Static" for both the evolutionary and differentiable paradigms. Since Random Static trains static architectures while EquiNAS$_E$ and EquiNAS$_D$ dynamically search for both architectures and parameters, we take the best 5 architectures for each and retrain their param-

eters from scratch as in the standard NAS paradigm, labeled as EquiNAS$_E$ Retrain and EquiNAS$_D$ Retrain, respectively, for fair comparison to Random Static.

The results of these additional experiments are compared against those of our main algorithms and baselines in Figures 5 and 8.

Shown in Figure 5, EquiNAS$_E$ outperforms EquiNAS$_E$ Random Select, showing the benefit of using informed selection to guide the relaxation of equivariance constraints over training. Additionally, EquiNAS$_E$ Retrain outperforms the Random Static baseline, showing that using compute in an informed search is more beneficial than just randomly searching the space of static architecture constraints.

EquiNAS$_D$ finds competitive architectures on average and can find architectures which outperform baseline choices like architectures fully equivariant to $C_1$ or $D_4$. From comparing EquiNAS$_D$ Retrain to EquiNAS$_D$ results in Figure 8, retraining a resulting architecture is not consistently better or worse than using the final weights from EquiNAS$_D$, showing that there isn't a disadvantage to training weights during search and avoiding the additional cost of a post-search training step.

The use of randomized loss information in Random Static in Figure 8 shows that an informed search for architecture hyperparameters is generally useful. However, experiments on the ISIC benchmark demonstrates that the architecture search can be deceptive and that random loss information can outperform informed loss. This motivates exploration into the use of noise during the search process for architectural parameters.

A sampling of random continuous architectures in Random Static in Figure 8 shows that random architectures can perform well on problems where fully equivariant architectures like the $D_4$ baseline already perform well. However, on the Galaxy10 problem, the $D_4$ and $C_4$ baseline have high variance, suggesting that a fully equivariant architecture is sub-optimal. On this baseline, EquiNAS$_D$ greatly outperforms a search of random architectures, demonstrating that EquiNAS$_D$ can discover the appropriate equivariance for a specific dataset over fixed or randomly selected architectures.

The search space for EquiNAS$_D$ is already well-formed for random architectures, compared to the discrete search space of EquiNAS$_E$. This is enabled by the $[G]$-mixed equivariant layer, which is a contribution of this work. Random non-mixed equivariant architectures do worse on all three benchmarks compared to random architectures which use the $[G]$-mixed equivariant layer. This can explain why the EquiNAS$_D$ results are closer to random baselines than the EquiNAS$_E$ results, as the search space permits easily finding the appropriate mix of equivariances compared to a discretized search space.

Our methods enable searching for both architecture and parameters concurrently in a single training process. This approach is more efficient than NAS approaches that only search for architectures, requiring a retraining process within the resulting architecture for evaluation. However, these ablation and random search comparisons show that our algorithms may get performance gains from adding a retraining phase with a tradeoff of further compute cost.

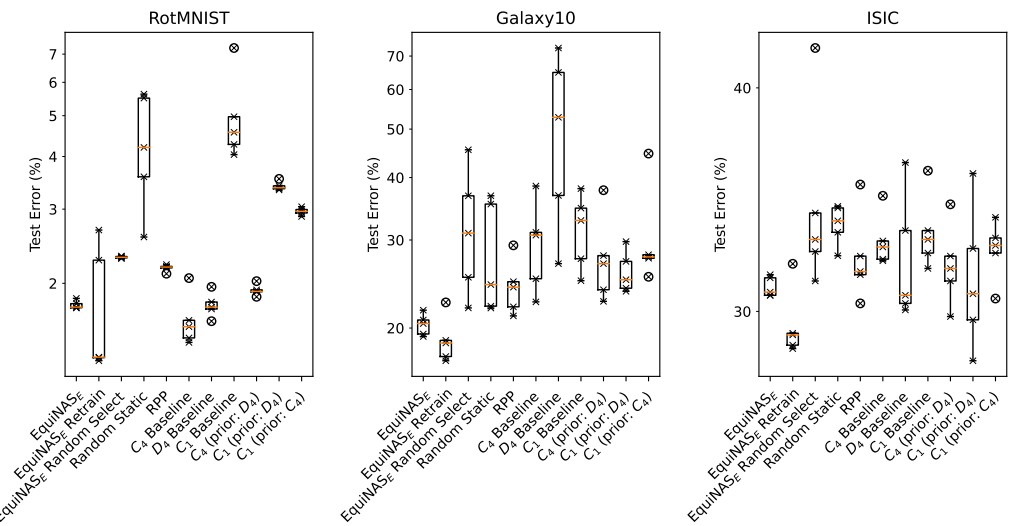

Figure 5: Test errors for experiments of Section 6.1.

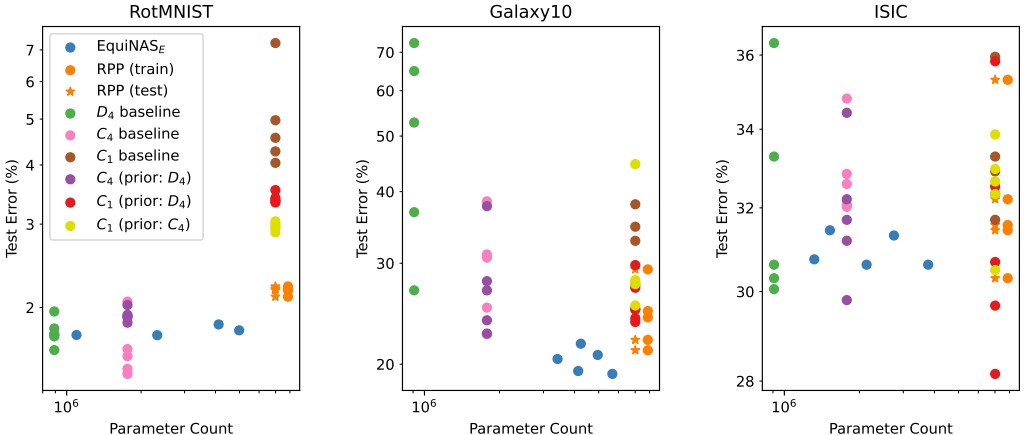

Figure 6: Test errors against stored parameter count for experiments of Section 6.1. For EquiNAS$_E$, parameter counts of the final models are shown, although training begins with the same parameter count as the $D_4$ baselines as shown in Figures 2 and 7. Although RPP trains with a higher parameter count, the parameters may be stored in a single filter per layer using Equation 11 for testing.

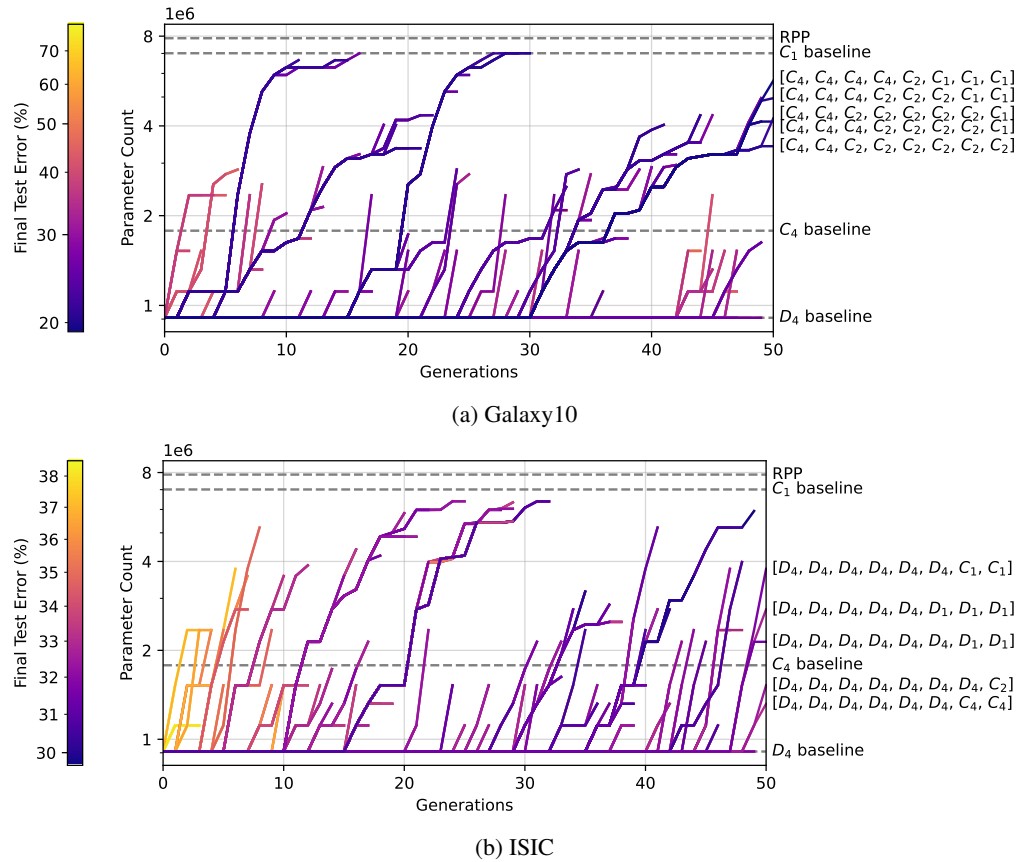

(a) Galaxy10

(b) ISIC

Figure 7: Historical parameter counts of each selected individual for EquiNAS$_E$ (see Figure 2 for RotMNIST).

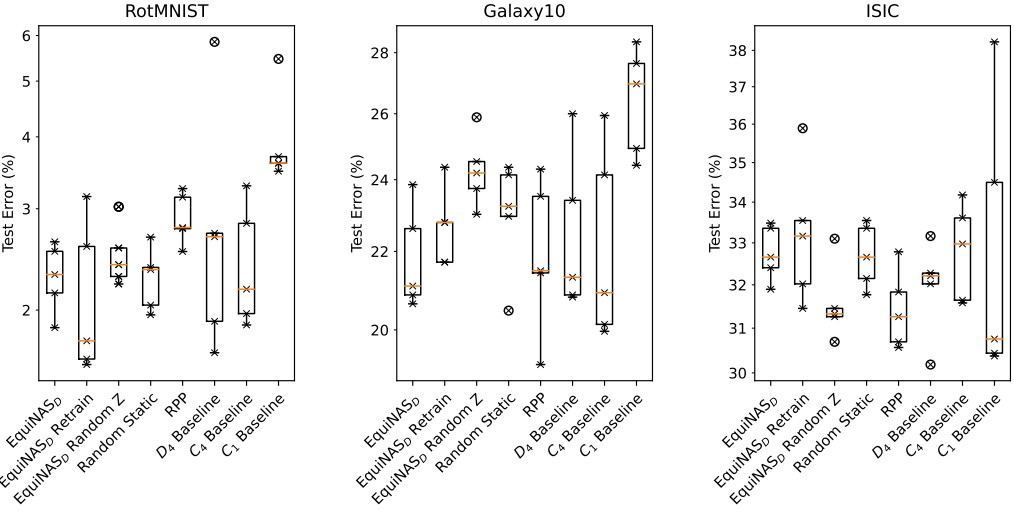

Figure 8: Test errors for experiments of Section 6.2.

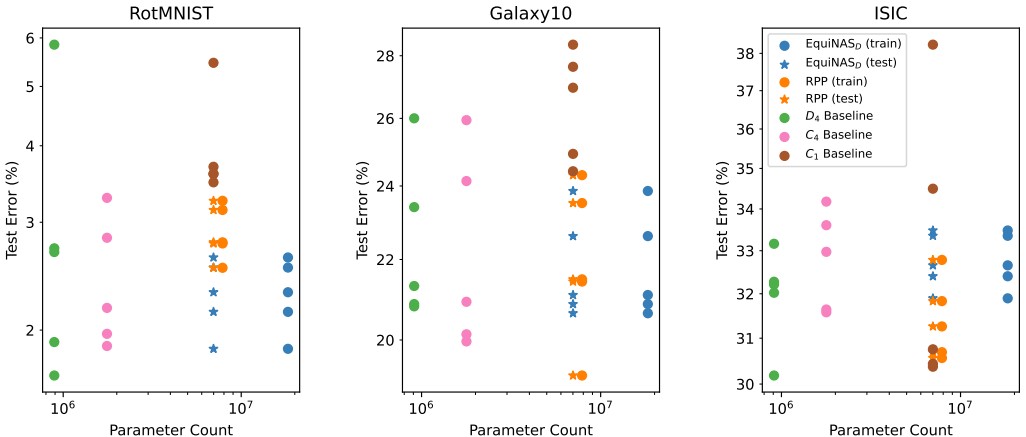

Figure 9: Test errors against stored parameter count for experiments of Section 6.2. Although EquiNAS$_D$ and RPP train with a higher parameter count, the parameters may be stored in a single filter per layer using Equation 11 for testing.

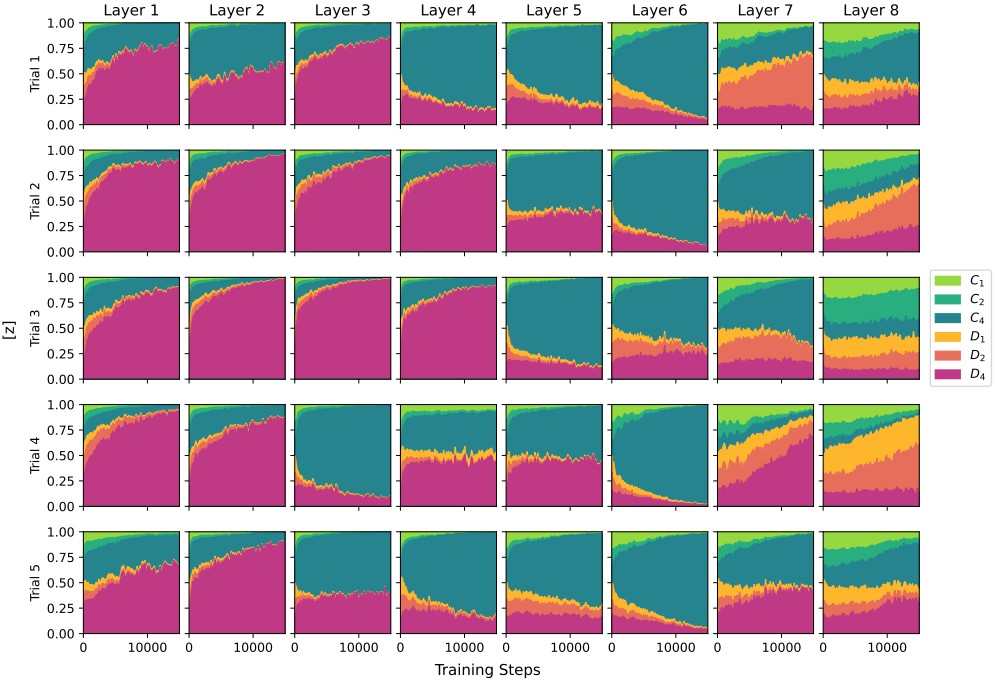

Figure 10: Architecture weighting parameters by layer for all trials on RotMNIST.

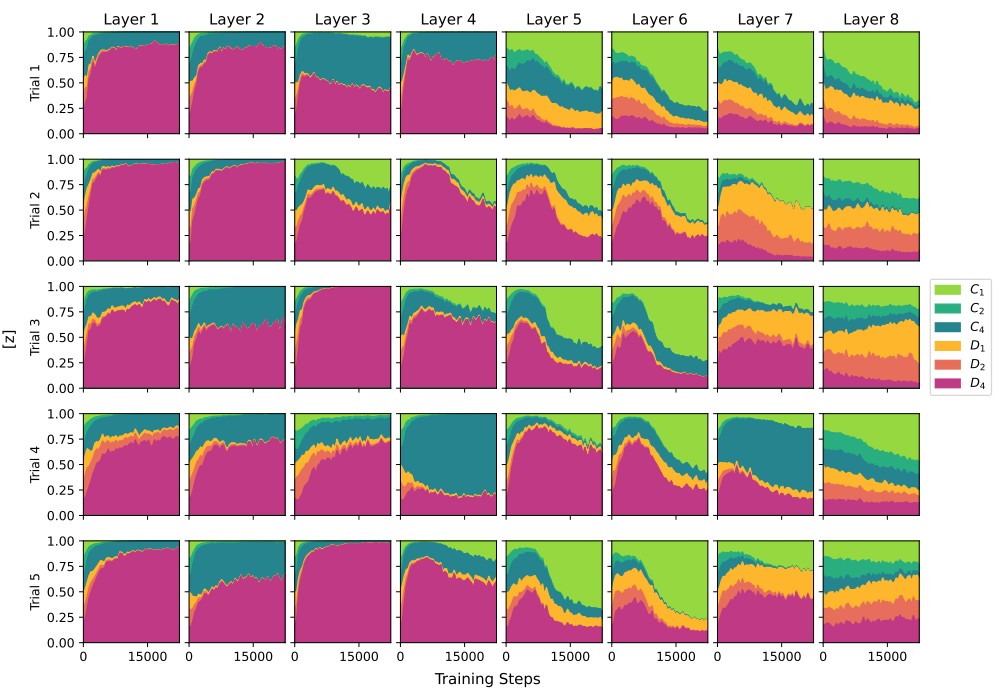

Figure 11: Architecture weighting parameters by layer for all trials on Galaxy10.

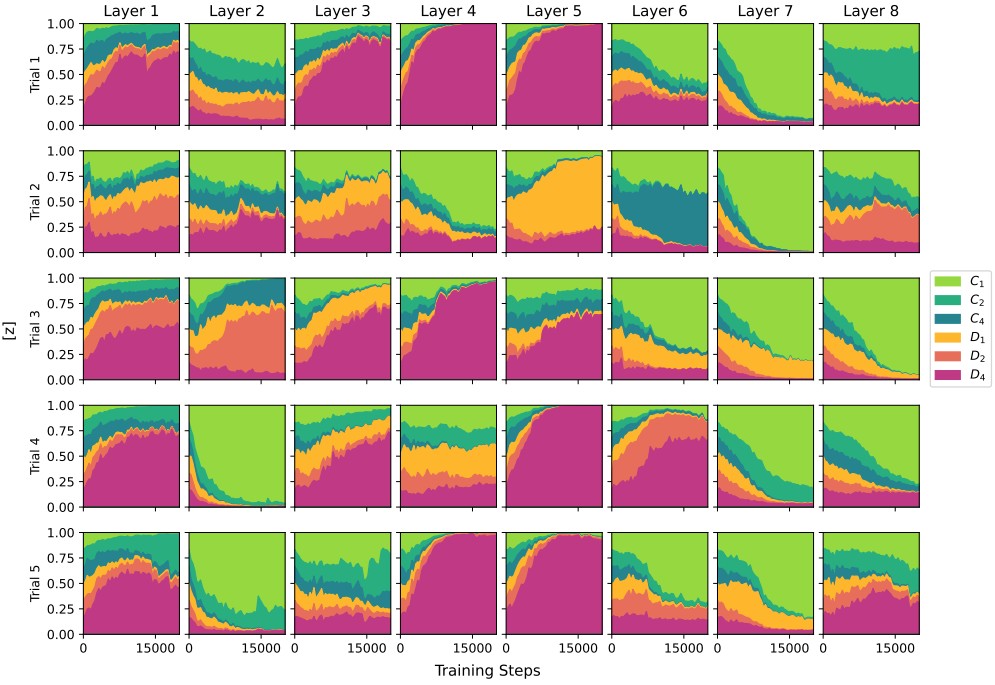

Figure 12: Architecture weighting parameters by layer for all trials on ISIC.

