# OpenReview forum: "Equivariance-aware Architectural Optimization of Neural Networks"
_ICLR.cc/2023/Conference — ICLR 2023 poster_

### Official Review · Reviewer_W8Dc · 2022-10-24

**Confidence:** 3
**Clarity, Quality, Novelty And Reproducibility:** The paper is mostly well-written and …
**Correctness:** 3
**Technical Novelty And Significance:** 3
**Empirical Novelty And Significance:** 3
**Recommendation:** 6

**Strength And Weaknesses:**

**Strength**
1. The paper is well motivated. Most G-CNN literature assume that the symmetry group of the learning task is explicitly known and perfect. The paper instead uses a equivariance-aware NAS procedure to search for the optimal architecture for (potentially) unknown and imperfect group transformations.
2. The paper is mostly well-written.
3. The experiments are detailed and thorough.

**Weakness**
1. The equivariance relaxation morphism, which relies on reparameterizing the group convolution on a subgroup, seems to be limited to discrete group. Can this be extended to continuous groups?
2. Moreover, the relaxation morphism is also restrictive in the sense that the number of the unstructured channel (c_out) cannot be adjusted after changing to a subgroup. This inevitably increases the model size as group size decreases, which is indeed the case for EquiNAS_D.
3. The clarity of the paper in section 5.1 can benefit from more explanation on the technical terms such as population, pareto selection, pareto front, etc.
4. In MNIST_rot, the review would like to see what happens when $C_8$ and $D_8$ are included in the picture.
5. If I understand correctly, the reported EquiNAS_E result on Galaxy10 in Table 1 might be misleading. Figure 4(a) shows that only one NAS output in the end achieves significantly smaller test error compared to all the remaining results.
6. Are the models involved in EquiNAS_D significantly larger compared to the baseline? After all, a linear combination of layers equivariant to different groups is taken. Does that mean the model is slow to train and (especially) test?

**Summary Of The Paper:**

This paper proposes two methods for equivariance-aware neural architectural search (NAS). The motivation for this work stems from the fact that symmetries present in a data set might often be imperfect or non-explicitly known. The first method is based on the proposed equivariance relaxation morphism, a procedure that equivalently reparameterizes a group convolution layer to operate with equivariance constraints on a subgroup; an evolutionary NAS based on such morphism is later adopted. The other method, named $[G]$-mixed equivariant layer, linearly combines layers equivariant to different groups, whose coefficient is search through a differentiable NAS. Thoughtful experiments on three datasets are conducted to demonstrate the performance.

**Summary Of The Review:**

I think the paper is interesting. I am willing to adjust my rating based on the authors' response to my questions.

---

> ### Author Response · Authors · 2022-11-08
> **Response to review from W8Dc**
>
> Thank you for your thorough review. We appreciate your noted strengths, such as the paper being well-written and the experiments being detailed and thorough, as well as your constructive comments, which we respond to individually as follows.
>
> Responding to “Weakness”:
>
> 1. Our methods should extend to continuous groups that are represented discretely, like in steerable convolutional layers, as long as G’ \G is finite. We have added this extension proposal to the paper in Section 7.
> 2. The equivariance relaxation morphism is just one atomic morphism that essentially transforms group element channels into less structured channels, such that the shape (and values) of the expanded filter remain the same. If widening the layer is desired, this could be done by applying other methods such as width morphing as described in Wei et al. (2016). However, reducing the channel count would necessarily change the function of the layer. The most basic way to preserve parameter count while changing equivariance is to use the same learnable parameters with parametrized equivariance to a less expressive group (i.e. expanding the same parameters with group actions of elements of $C_2$ instead of $C_4$ such that the expanded filter has half as many incoming and outgoing channels), but this would only hurt the network by reducing its capacity.
> 3. We have added some additional information on terminology used in Section 5.1 into Appendix A.2.
> 4. $C_8$ and $D_8$ require interpolation, so the relaxation does not exactly hold our claimed properties in this case. We had included these groups in early experimentation, but the lack of exact symmetry makes implementing the necessary indexing highly complex and therefore infeasible within the rebuttal window in our current codebase.
> 5. We found a bug in how the Galaxy10 dataset was being loaded, so the experiments were rerun after fixing the bug and the results have been updated. Our new results are more consistent and clear.
> 6. By parameter count, their size scales linearly by the size of $[G]$. However, using the trick shown in Equation 11, each [G]-mixed equivariant layer can be evaluated by a single $G’$-equivariant convolutional layer in the forward pass. Thus, for evaluation and inference, all models in our experiments can be computed and stored with the same parameter count as the $C_1$ baseline. This has been detailed in the updated version, with additional plots comparing parameter counts in Appendix D, although these counts are more relevant to storage than to computational costs, as for all models in our experiments, the filters are expanded to the same size for evaluation. Also, although EquiNAS$_D$ are more expensive to train than static architectures, they enable architecture search to avoid a bunch of trial-and-error runs to manually optimize the architecture.
>
> We appreciate any further feedback you have on our current and planned changes.

---

> > ### Comment · Reviewer_W8Dc · 2022-12-06
> > **Thank you for your response**
> >
> > I really appreciate the authors' clear response. Most of my concerns have been addressed, and the fixed experiment on Galaxy10 is convincing. I am raising my rating to marginally above acceptance.

---

### Official Review · Reviewer_6SKq · 2022-10-24

**Confidence:** 3
**Correctness:** 3
**Technical Novelty And Significance:** 3
**Empirical Novelty And Significance:** 3
**Recommendation:** 6

**Clarity, Quality, Novelty And Reproducibility:**

The paper is fairly clear with some weak spots corresponding to my questions above.  It could use illustrations and examples to help explain the method.  The quality is okay.  The experiments could be more thorough but do provide a basic trial for the proposed method.  The method is tested on 3 different datasets. The method seems novel to me.  There is no code provided but it is promised.  The description seems clear enough to repeat similar experiments.

**Strength And Weaknesses:**

## Strengths
- Equivariant networks have more hyperparameters to tune due to the choice of symmetry group.  For example, in E(2)-CNN (https://arxiv.org/abs/1911.08251), the authors experiment with symmetry groups $C_n$ and $D_n$ for many values of n before finding n=12 to work best.  This paper proposes a method for automatically determining the correct level of symmetry.
- Moreover, on many problems end-to-end strict symmetry is not desirable if the symmetry in the domain is only approximate or if symmetry constraints interfere with optimization.  The proposed method adds to the relatively few methods which learn relaxed symmetry constraints and adapt to the level of symmetry in the data.
- I found the equivariance relaxation morphism to be a simple and effective idea for relaxing symmetry constraints in networks on a per-layer basis.
- The [G]-mixed equivariant layer also seems like a reasonable idea for selecting the degree of equivariance by differentiably choosing weights over kernels with different levels of equivariance.  In theory this allows the model to learn a specific equivariance or an approximate equivariance which is in-between.  It also allows the network to reduce the symmetry constraint in later layers, which is what is done effectively by CNNs in practice by downsampling with stride.
- Experimental results on the learned subgroup weights (Fig. 2) reveal interesting trends showing that the networks do select for higher equivariance in early layers and lower equivariance in later layers which matches what has found to be effective in practice for equivariant networks.


## Weaknesses / Questions
- I am not sure of the framing of the method.  I am not an expert on NAS and am open to correction, but it seems to me both proposed methods find a trained network, not an architecture, since they are training the network weights and selecting an architecture at the same time.  The final architecture may not have great performance if trained from scratch.  In this sense, it seems the method is more a special training procedure than NAS method.   EquiNAS_D, in particular, is simply an architecture in which some weights are optimized alternatingly.
- Given the evolutionary algorithm starts with a single network, how do we know to what extent its evolution is guided by architecture versus the specific weights in the mutated networks?  In experiments, I think both population level variance and initialization variance should be accounted for.
- One of the biggest issues is the fact that the relaxation morphism increases the number of parameters making it difficult to distinguish performance increases are from relaxing constrains or increasing parameter count.  The authors do discuss this issue and suggest their optimization strategy over both parameter count and accuracy helps.  It would be better, however, if the number of parameters could be preserved by the relaxation.
- The experimental results are not completely convincing of when this method would add significant value over simply doing hyperparameter search over equivariant networks.  In Table 1, an equivariant baseline is best on rotMNIST and ISIC, and a non-rotationally equivariant network with equivariant initialization is best for Galaxy10.  In table 2, EquiNAS_D does do best for rotMNIST, but the score is much lower than the best values in E(2)-CNN after full tuning of the best equivariant method.  In Galaxy10, EquiNAS_D does outperform, but with very high variance.
- I have some issues with the clarity of the description of 4.2.  The condition that all groups in $[G]$ be subgroups or supergroups of all other groups in $[G]$ does not appear to be met by the expermental example since $D_2$ is not a subgroup or supergroup of $C_4$.  It's also not clear to me why this condition is necessary.  It is a fairly strict condition, although I think it would include useful cases.  I am also a bit unclear on the form of the input signal $f$.  In order to be convolved with $\tilde{\Psi}$ and have $G$-equivariance for each $G$ in $[G]$, $f$ should be defined over a supergroup of all $G \in [G]$.  Is this the case?   I am correct that if $z_G$ is a one-hot vector on the group $G$, then the operation is $G$-equivariant?

## Questions
- If the evolution algorithm can only mutate in the direction of symmetry constrain relaxation, doesn't this bias the algorithm towards relaxing all the symmetry constraints?
- Does this work have a relation to diffstride (https://arxiv.org/abs/2202.01653) in that stride is typically a hyperparameter that corresonds to a choice of the subgroup of the translation group and can be shown to differentiably optimized?
- After equivariance is broken in a given layer, there isn't much point in principle to persevering it downstream.  Are mutations automatically applied to all following layers or did your method discover this?  If so, I think that is interesting enough to be considered a strength.

## Minor
- In Eqn 1, if you reverse the order of $f$ and $\Psi$ and drop the sum over c and indices $c,d$, the equation can be simplified using matrix multiplication.
- 4.1, para 2, "neutral element" -> "identity element"
- 4.1 may benefit from writing as a proposition
- 4.1, Para 3, Line 5,  $h \in G'$ -> $h' \in G'$
- Sec 5., Line 4, typo "optimizffe"


**Summary Of The Paper:**

This paper proposes two approaches to optimizing the weights and architecture of networks with approximate equivariance.  The first approach EquiNAS_E uses an evolutionary algorithm to progessively weaken the symmetry constraints in the layers of an equivariant NN as it trains.  The second EquiNAS_D uses differentiable weights, optimized alternatingly with the normal network weights, which combine different kernel with different levels of symmetry constraints.  Both methods are tested on image classification tasks.

**Summary Of The Review:**

This method seems quite interesting to me due to the fact that there are datasets in which strict equivariance may not perform well due to being too strict of an inductive bias or because the constraints interfere with optimization.  Moreover, equivariant networks have a great deal of hyperparameters in the amount of symmetry they exhibit at various levels, so a more sophisticated approach than random searching is worthwhile.  The proposed idea for relaxing symmetry constraints is appealingly simple and reasonable.  However, the experimental results are not that convincing as the proposed method is often outperformed by equivariant networks or by relaxed networks with equivariant initialization.  In the case EquiNAS_D does significantly outperform on Galaxy10, it has very high variance.  Since the relaxation method also adds parameters, it is not clear when improved performance is due to parameter count vs. too strong symmetry constraints.

---

> ### Author Response · Authors · 2022-11-08
> **Response to review from 6SKq**
>
> Thank you for your thoughtful and thorough review. We appreciate your detailed noted strengths as well as constructive comments to improve the work, which we respond to individually below.
>
> Responses to “Weaknesses / Questions”
> 1. Our methods optimize both the architecture and its parameters in a single process, which is different from many NAS works but still a defined NAS paradigm (see Definition 8 in Lindauer & Hutter (2020))
> 2. There are many sources of variability in NAS, not only inherent in a NAS algorithm itself but also in other components, such as the learning algorithm used. You seem to suggest using the same initialization but with different seeds for the training process, but this is not standard practice as handling all sources of variability individually would quickly become a very complex and expensive problem.  Could you clarify this point?
> 3. Such a relaxation that preserves parameter count is not possible without changing the function of the layer, thus breaking the morphism definition. The most basic way to accomplish what you are suggesting is to use the same learnable parameters with parametrized equivariance to a less expressive group (i.e. expanding the same parameters with group actions of elements of $C_2$ instead of $C_4$ such that the expanded filter has half as many incoming and outgoing channels), but this would only hurt the network by reducing its capacity.  We have added plots that compare final accuracy against train and test parameter counts in Appendix D, although these counts are more relevant to storage than to computational costs, as for all models in our experiments, the filters are expanded to the same size for evaluation.
> 4. We believe our updated results ameliorate these concerns, with clearer distinctions between our works and baselines. Also, this point ends in an incomplete sentence: do you have anything to add?
> 5. This is a good point regarding the condition. This requirement was from an older version of our work, but we have since relaxed this constraint and ended up experimenting with a group lattice. We relaxed this condition in the paper. $f$ is necessarily defined over subgroup $G’$ of all groups in $[G]$, if it is itself output from a $[G]$-mixed equivariant layer: note that it is convolved with respect to group $G’$ in Equations 10 and 11. Your final sentence is correct: this is actually how we implemented our codebase.
>
> Responses to “Questions”
> 1. You are correct that the equivariance relaxation morphism only works in one direction, thus EquiNAS$_E$ necessarily can only search in one direction. There is not a complementary morphism that can go in the other direction without changing the function of the layer, although determining when and how to do this with some functional change, possibly with regularization and projection, is an interesting future study we hope to work on.
> 2. We are very interested in works like this that permit gradient-based optimization, as differentiable NAS approaches. However, striding is quite distinct from group equivariance: we don't have an intuition on how groups could be “relaxed” as can be done for striding or other aspects like kernel size (Romero et al., 2022), although steerable convolutional layers also use the fourier transform (Weiler & Cesa, 2019). We plan to extend our work to the steerable case in future work.
> 3. We do restrict the offspring generation to only relax a layer if no layer’s parametrized equivariance is not a supergroup of that of all following layers, inspired by works such as Weiler & Cesa (2019). Relaxing this restriction, as done by Romero & Lohit (2022), would be an interesting study but would greatly widen the search space, thus being considerably more expensive to search.
>
> Regarding your response in “Clarity, Quality, Novelty And Reproducibility”, we have added illustrative figures for each mechanism.
>
> We appreciate any further feedback you have on our current and planned changes.
>
>
> References:
> * Marius Lindauer and Frank Hutter. Best practices for scientific research on neural architecture search. Journal of Machine Learning Research, 21.243:1-18, 2020.
> * David W Romero, Robert-Jan Bruintjes, Jakub Mikolaj Tomczak, Erik J Bekkers, Mark Hoogendoorn, Jan van Gemert. FlexConv: continuous kernel convolutions with differentiable kernel sizes. In International Conference on Learning Representations, 2022
> * David W Romero and Suhas Lohit. Learning partial equivariances from data. Advances in Neural Information Processing Systems, 35, 2022.
> * Maurice Weiler and Gabriele Cesa. General E(2)-equivariant steerable CNNs. Advances in Neural Information Processing Systems, 32, 2019.

---

> > ### Comment · Reviewer_6SKq · 2022-12-05
> > **Thanks for Response**
> >
> > I have read the other reviews, author responses, and revised paper.  I appreciate the authors effort here.
> >
> > 2. I am interested in the extent to which the final architecture is determined by the specific initialization of the weights versus the task.  Thus the suggestion to consider these two sources of randomness.  However, I agree that understanding variance across all sources of randomness is the most critical aspect.
> > 3. In many cases architectures are selected which outperform more highly constrained architectures, but have more parameters.  How do we know if the performance increase is due the architecture or the parameter count?  A simple way might be to train, e.g., D_4-equivariant architecture with the same number of parameters as the best model found by EquiNAS.
> > 4. The experimental results are now indeed stronger.  I am still not fully convinced, however, of the ultimate usefulness of the method in applications, given that the best performing method does not outperform SoTA on e.g. RotMNIST.  Is there an example in the paper where the proposed method outperforms all competing equivariant methods (even including settings where relaxed equivariance could be an advantage)?
> >
> > Q2. I don't think that striding is different from group equivariance; I think it is quite analogous to the subgroup symmetry considered here.  Strided convolution enforces equivariance to a subgroup $d\mathbb{Z}^2$ of the translation group $\mathbb{Z}^2$ where the group action on the input is a shift by $d$ pixels and by $1$ pixel on the input.
> >
> > Overall, while I still have some lingering doubts, I think the paper has improved (Figure 1 is nice), the experimental results are better, and the idea is interesting and worthwhile.  In other words, I believe the strengths outweigh the weaknesses and am in favor of acceptance.

---

### Official Review · Reviewer_RfYJ · 2022-10-27

**Confidence:** 3
**Correctness:** 4
**Technical Novelty And Significance:** 3
**Empirical Novelty And Significance:** 2
**Recommendation:** 6

**Clarity, Quality, Novelty And Reproducibility:**

Clarity: the writing is quite clear.
Quality: the empirical evaluation is somewhat limited in scope.
Novelty: the idea of searching over equivariant groups is new and interesting.
Reproducibility: code is promised upon publication.

**Strength And Weaknesses:**

### Strengths:
1. The problem of finding the correct equivariance that can exploit symmetries in the data is important for the development of general-purpose NAS methods.
2. The proposed approaches are simple and natural procedures.
3. The writing is fairly clear and the experimental results are interesting.
4. Code will be released upon acceptance.

### Weaknesses:
1. The experimental evaluation is somewhat limited. Given the fairly direct extension of prior work (morphisms and weight-sharing) to actually developing the search algorithms, I would expect more emphasis to be placed on experimental depth. This could either come in the form of evaluation on interesting benchmarks where we might expect finding symmetries to outperform regular CNNs (e.g. Tu et al. (2022)) and/or more detailed analysis on more involved search spaces of whether the proposed methods are recovering the “right” symmetry for the dataset, even if synthetic.
2. The authors do not consider the random search baseline for their method. More generally, it would be useful to follow the NAS reproducibility checklist here (Lindauer & Hutter, 2020).

### References:
- Lindauer, Hutter. Best practices for scientific research on neural architecture search. JMLR 2020.
- Tu, Roberts, Khodak, Shen, Sala, Talwalkar. NAS-Bench-360: Benchmarking neural architecture search on diverse tasks. NeurIPS 2022.

**Summary Of The Paper:**

This paper proposes to search for neural architectures whose layers have equivariance that can exploit symmetries in the data. They propose two methods—network morphisms combined with evolutionary search and mixture relaxations combined with differentiable NAS—to search the space of symmetries. The methods are evaluated on some image classification tasks.

**Summary Of The Review:**

I am currently leaning against acceptance to the rather limited empirical evaluation.

# Post-rebuttal

Given the additional evaluations, with new datasets and comparisons with baselines, I think the paper has sufficient experimental backing / interesting methodology to be accepted.

---

> ### Author Response · Authors · 2022-11-08
> **Response to review from RfYJ**
>
>
> Thank you for your review, including notable strengths (such as the importance of the tackled problem and interesting current results) and constructive weaknesses that we respond to below.
>
> Responses to “Weaknesses”:
> 1. NAS-Bench-360 is a very interesting benchmark that we hope to use in future work since it spans a breadth of tasks, data structures, and symmetries, but we do not think it is feasible to implement within the rebuttal period: not only in integrating their software and our codebase but also designing the architectural search space for each tasks. We hope our currently improved results and other proposed experiment extensions are sufficient to support our claims. However, we are currently implementing various combinations training on MNIST without augmentation then validating and/or testing on MNIST with 180 degree rotation augmentation to explore out-of-distribution properties and discovery of $C_2$ from the subgroup lattice we test.
> 2. We did follow the relevant parts of the NAS checklist, although many aspects were out of scope when the focus of our work is the new mechanisms that potentiate NAS. Properly designing random searches that are comparable to our NAS algorithms is rather tricky, since we are in the AutoML paradigm of NAS (see “Definition 8 (AutoML variant of NAS)” of the checklist) where both the architecture and parameters are optimized in a single process. Usually NAS methods can compare to random search methods that spend the same compute time on training randomly generated architectures, but this is not as applicable for EquiNAS$_E$, since this would translate to randomly generating 5 architectures and training them for the same time, nor for EquiNAS$_D$, since the architecture is not finally discretized as in other differentiable NAS works. We propose the following random search baselines that are relevant to our methods. We are currently implementing them to include their results before the end of the rebuttal period, but please let us know if you have something else in mind:
>     * EquiNAS$_E$: replace the pareto-front parent selection with random selection from the current population.
>     * EquiNAS$_D$: replace the gradient descent update of architecture weighting parameters with random updates.
>
> We appreciate any further feedback you have on our current and planned changes.

---

### Official Review · Reviewer_7fUJ · 2022-10-31

**Confidence:** 3
**Correctness:** 4
**Technical Novelty And Significance:** 3
**Empirical Novelty And Significance:** 3
**Recommendation:** 6

**Clarity, Quality, Novelty And Reproducibility:**

Quality and Novelty - The paper tackles an important problem and has a reasonable amount of novelty as described in previous section.

Clarity - While the paper is understandable for someone who has the necessary group theory background - it is almost impossible for general audience - would recommend the authors add necessary background and preliminaries to the appendix.

Reproducibility - Currently code not available

**Strength And Weaknesses:**

**Strengths:**
1. The formulation of partial/ approximate equivariances/ invariances $-$ as a mixture over subgroups is interesting and an important problem to tackle.
2. Strong experimental results for both their learning paradigms.
3. For a person with sufficient group theory background - the paper is easy to understand.

**Weaknesses and correspnding questions/ suggestions**:
1. Lack of comparison of the evolutionary paradigm with a similar approach [1] which searches over the subspace lattice  to identify te groups to be equivariant/ invariant to.
2. It is hard to see how this extends to non-finite groups which are not compact. Would suggest that the authors comment on this.
3. While the paper is understandable for someone who has the necessary group theory background - it is almost impossible for general audience - would recommend the authors add necessary background and preliminaries to the appendix.
4. Studies in out of distribution settings are missing (for e.g. train only contains images rotated by 0-15 degrees, but test has those between 60-180 degrees) - this is will give us more insight into the weights learned.

**Minor:**
1. Please make sure the title on openreview matches the title on the paper

**References**
1. Mouli, S. Chandra, and Bruno Ribeiro. "Neural Networks for Learning Counterfactual G-Invariances from Single Environments." ICLR 2021

**Summary Of The Paper:**

In this work, the authors explore technique towards leveraging equivariances to subgroups rather than the entire group. First they propose equivariance relaxation morphism $-$ which partially removes weight sharing constraints on the entire group (without losing out on functionality) and the mixed layer allows to learn partial equivariance as a weighted sum. The authors then propose two equivariance aware NAS algorithms - one evolutionary and the other differentiable - and show good experimental results in both cases.

**Summary Of The Review:**

Currently the strengths of the paper out weigh the weaknesses. I would be happy to increase my scores if the authors can answer the raised questions.

---

> ### Author Response · Authors · 2022-11-08
> **Repsonse to review from 7fUj**
>
> Thank you for your thorough review of our work. We appreciate the strengths you have noted, such as how important this problem is and already strong experimental results.
>
> Responses to “Weaknesses and corresponding questions/ suggestions”:
> 1. We added a comparison and citation to Mouli and Ribeiro (2021). They do use a similar equivariance relaxation mechanism, but rather for constructing a regularization term towards equivariance compared to reparametrizing the layer. Their use case is also different, intended for one-shot environment extrapolation, although this paradigm has useful benefits in general learning for generalization properties.
> 2. We conjecture that our methods could extend to continuous groups that are represented discretely, such as in steerable convolutional layers, as long as G’ \G is finite. We have added this extension proposal to the paper in Section 7.
> 3. We have added an appendix that gives relevant background information on group theory. The visualizations we added in FIgure 1 should also help the audience understand what each mechanism is doing, even if they don’t have a strong group theory background.
> 4. We are currently implementing various combinations of training on MNIST without augmentation then validating and/or testing on MNIST with 180 degree rotation augmentation to explore out-of-distribution properties and discovery of $C_2$ from the subgroup lattice we test.
>
> Re: “Minor”: thank you for pointing out that the OpenReview title and paper title did not match: the previous OpenReview title was a placeholder for abstract registration, and we were not aware that our attempt to change it to the paper title was not successful. The Author Guide suggests that changing the OpenReview title should still be possible, but we do not see this option with the Rebuttal Revision. We have changed the paper title to match that of OpenReview for now, but desire to finally change the title to “Equivariance-aware Architectural Optimization of Neural Networks”.
>
> We appreciate any further feedback you have on our current and planned changes.

---

> > ### Comment · Reviewer_7fUJ · 2022-11-28
> > **Response to the authors**
> >
> > Thank you very much for the clarifications and the updates to the manuscript. After reading through the other reviews (for now) I would like to keep my score, but will increase to an 8 if the new proposed MNIST experiments showcase the utility in out of distribution settings as well.

---

> > > ### Author Response · Authors · 2022-11-30
> > > **Post-rebuttal response to Reviewer 7fUJ**
> > >
> > > Please refer to our general comment "Rebuttal Update" above as well as the updated PDF to see the changes we ended up making during the rebuttal period. In particular, we changed the proposed MNIST experiments to instead explore explicitly added symmetries, as well as added further experiments for ablation of "smart" search strategies, random architecture search, and retraining of searched architectures. Our methods do not (nor are not intended to) directly handle out of distribution settings, as they optimize statistics over the training set without explicit regularization towards more general equivariances; however, we note that such regularization could easily be added in order to more closely follow the ideas of Mouli and Ribeiro (2021). Another interesting follow-up experiment would be to apply the retraining protocol (which is the standard NAS paradigm, while our base algorithms do not require retraining parameters within the searched architecture) to the out-of-distribution setting, retraining the architectures on different data distributions. However, we can no longer update the PDF to include such experiments.

---

### Author Response · Authors · 2022-11-08
**Prelimininary general response to reviews**


We thank all reviewers for their thorough and constructive responses. We appreciate the strengths pointed out, such as the simplicity of the relaxation (RfYJ and 6SKq) and the experiments and results supporting our contributions (7fUJ, RfYJ and W8Dc). As for weaknesses and requested changes, we have noted large changes in this general response below, with smaller changes and discussions noted in personal responses to each reviewer. In our updated PDF, we have highlighted changes in red. We plan on doing a second update by the end of next week with further experimental results but hope to start the discussion with this response.

We apologize that the OpenReview title and paper title did not match (as noted by 7fUJ): the previous OpenReview title was a placeholder for abstract registration, and we were not aware that our attempt to change it to the paper title was not successful. The Author Guide suggests that changing the OpenReview title should still be possible, but we do not see this option with the Rebuttal Revision. We have changed the paper title to match that of OpenReview for now, but desire to finally change the title to “Equivariance-aware Architectural Optimization of Neural Networks”.

During the review period, we found and fixed a bug in how the Galaxy10 dataset was handled, so those results were updated. Additionally, ISIC trials in Section 6.1 were extended to 50 generations, with updated results. These new results and relevant discussion address concerns by reviewers 6SKq and W8Dc, although both of these reviewers as well as reviewer 7fUJ also pointed out strengths in the experimental protocol and results.

We added an illustrative figure for each of the two mechanisms as Figure 1, as requested by reviewer 6SKq. We also added figures that compare final accuracy against stored parameter counts at train and test time for all experiments in Appendix D.

We added an additional appendix section that gives an initial background on group theory (requested by reviewer 7fUJ) and on evolutionary method terminology (requested by reviewer W8Dc) in Appendix A.

We are currently working on the following experimental extensions. We expect to have these extensions by the end of the rebuttal period, unless any reviewers have further commentary on these plans.
* Training on MNIST without augmentation, testing on MNIST with 180 degree rotation augmentation (for all methods) - requested by reviewers 7fUJ and RfYJ.
* Random search: EquiNAS$_E$ with random parent selection (instead of pareto-front selection), EquiNAS$_D$ with random $Z$ updates (instead of gradient descent) - requested by reviewer RfYJ.

We welcome any further discussion and constructive comments.

---

### Author Response · Authors · 2022-11-19
**Rebuttal Update**

We have updated our work with new experiments, mostly in Appendix D. This is our final planned update.

We originally planned to explore how our methods work when the test set is out of the training distribution. However, our methods naturally adapt to the symmetries present in the training set, rather than generalizing to all possible symmetries. We thus decided to instead analyze how our methods adapt to symmetries in the training set by running EquiNAS$_D$ on 6 different augmentations of MNIST, comparing the architecture dynamics to study explicit versus inherent symmetries. This responds directly to a request by RfYJ.

We added two “flavors” of random search:
One already described that ablates smart architecture search: EquiNAS$_E$ Random Select with random parent selection (instead of pareto-front selection) and EquiNAS$_D$ Random Z with random $Z$ updates (instead of gradient descent) by shuffling $Z$ gradients.
One more akin to standard NAS random search: for the evolutionary paradigm, training 30 randomly selected static architectures in the discrete architecture search space (still with the constraint of less constrained layers deeper in the network) for the same training time and selecting the top 5 by validation accuracy, and for the differentiable paradigm, training 25 randomly selected static architectures in the continuous architecture search space and selecting the top 5 by validation accuracy. 30 and 25 were respectively calculated to be approximately the same compute cost as the trials of EquiNAS$_E$ and EquiNAS$_D$ presented in our results. These are both labeled as Random Static.

Since the Static Random trials train static architectures while EquiNAS$_E$ and EquiNAS$_D$ dynamically search for both architectures and parameters, we take the best 5 architectures for each and retrain their parameters from scratch in the standard NAS paradigm, labeled as EquiNAS$_E$ Retrain and EquiNAS$_D$ Retrain, respectively.

These results with analysis have been added to Appendix D, with some references to them added to Section 5.2 and the captions of Tables 1-2 as well as comprehensive discussion added in Section 7. We summarize the new findings below.

In the evolutionary paradigm, EquiNAS$_E$ outperforms EquiNAS$_E$ Random Select, showing the benefit of using informed selection to guide the relaxation of equivariance constraints over training. Additionally, EquiNAS$_E$ Retrain outperforms the Random Static baseline, showing that using compute in an informed search is more beneficial than just randomly searching the space of static architecture constraints.

In the differentiable paradigm, the results are less clear. EquiNAS$_D$ finds competitive architectures on average and can find architectures which outperform baseline choices like architectures fully equivariant to $C_1$ or $D_4$. From comparing EquiNAS$_D$ Retrain to EquiNAS$_D$, retraining a resulting architecture is not consistently better or worse than using the final weights from EquiNAS$_D$, showing that there isn't a disadvantage to training weights during search and avoiding the additional cost of a post-search training step. The use of randomized loss information in Random Static shows that an informed search for architecture hyperparameters is generally useful on tasks except ISIC, where random loss information outperformed informed loss. Random architectures seem to be able to perform well on problems where fully equivariant architectures like the $D_4$ baseline already perform well.

The search space for EquiNAS$_D$ is already well-formed for random architectures enabled by the $[G]$-mixed equivariant layer, compared to the discrete search space of EquiNAS$_E$. Random non-mixed equivariant architectures do worse on all three benchmarks compared to random architectures which use the $[G]$-mixed equivariant layer. This can explain why the EquiNAS$_D$ results are closer to random baselines than the EquiNAS$_E$ results, as the search space permits easily finding the appropriate mix of equivariances compared to a discretized search space.

Our methods enable searching for both architecture and parameters concurrently in a single training process. This approach is more efficient than NAS approaches that only search for architectures, requiring a retraining process within the resulting architecture for evaluation. However, these ablation and random search comparisons show that our algorithms may get performance gains from adding a retraining phase with a tradeoff of further compute cost.

---

### Decision · Program_Chairs · 2023-01-20

**Decision:**

Accept: poster

**Justification For Why Not Higher Score:**

The method has some merit in terms of idea and some proof of concept.
The practical benefit is still not fully convincible.
Both merit and practical benefit do not grant higher score.

**Justification For Why Not Lower Score:**

All reviewers in concensus for the merit of the partial and mixture equivariance and argue for acceptance.

**Metareview: Summary, Strengths And Weaknesses:**

We recommend accepting this paper to ICLR 2023.
Initially there were 3 positive and 1 negative reviewers. The negative reviewer updated their score to a positive one during the discussions after considering rebuttal and revision.
The paper's revision convinced the reviewers this work can be published in its current state. However, we do encourage the authors to consider/discuss remaining issues: (1) improvements factoring out number of parameters; and (2) practicality for real datasets.

**Note From Pc:**

if the above contains the word "oral" or "spotlight" please see: "oral" presentation means -> notable-top-5% and "spotlight" means -> notable-top-25%. As stated in our emails, we are disassociating presentation type from AC recommendations